# Morphological, Molecular, and Nutritional Characterisation of the Globe Artichoke Landrace “Carciofo Ortano”

**DOI:** 10.3390/plants12091844

**Published:** 2023-04-29

**Authors:** Enrica Alicandri, Anna Rita Paolacci, Giulio Catarcione, Alberto Del Lungo, Valentina Iacoponi, Francesco Pati, Giuseppe Scarascia Mugnozza, Mario Ciaffi

**Affiliations:** 1Dipartimento per la Innovazione nei Sistemi Biologici, Agroalimentari e Forestali (DIBAF), Università degli Studi della Tuscia, Via San Camillo de Lellis, 01100 Viterbo, Italy; 2Agenzia Regionale per lo Sviluppo e l’Innovazione dell’Agricoltura del Lazio (ARSIAL), Via Rodolfo Lanciani, 38, 00162 Rome, Italy

**Keywords:** *Cynara cardunculus* L. var. *scolymus*, genetic diversity, molecular markers, morphological traits, Folin–Ciocalteu assay, flavonoid content, nutritional values

## Abstract

The present study focused on the molecular, morphological, and nutritional characterisation of a globe artichoke landrace at risk of genetic erosion still cultivated in the municipality of Orte (Lazio Region, Central Italy) and therefore named “Carciofo Ortano”. Molecular analysis based on SSR and ISSR markers was carried out on 73 genotypes selected at random from 20 smallholdings located in the Orte countryside and 17 accessions of landraces/clones belonging to the main varietal types cultivated in Italy. The results confirmed that “Carciofo Ortano” belongs to the “Romanesco” varietal typology and revealed the presence within the landrace of two distinct genetic populations named Orte 1 and Orte 2. Despite the high level of within-population genetic variation detected, the two populations were genetically differentiated from each other and from the landraces/clones of the main varietal types cultivated in Italy. Morphological and nutritional characterisation was performed on representative genotypes for each of the two populations of the “Carciofo Ortano” and the four landraces/clones included in the varietal platform of the PGI “CARCIOFO ROMANESCO DEL LAZIO” used as reference genotypes (“Campagnano”, “Castellammare”, “C3”, and “Grato 1”). Principal component analysis showed that, of the 43 morphological descriptors considered, 12, including plant height, head shape index, head yield, and earliness, allowed a clear grouping of genotypes, distinguishing Orte 1 and Orte 2 populations from the reference genotypes. Regarding the nutritional composition of heads, particular attention should be devoted to the Orte 2 genotypes for their high dietary fibre, inulin, flavonoid, and phenol content, a feature that could be highly appreciated by the market.

## 1. Introduction

*Cynara cardunculus* L. var. *scolymus* (L.) Fiori (*2n* = *2x* = 34), commonly known as globe artichoke, is a perennial herbaceous open-pollinated species native to the Mediterranean basin that belongs to the Asteraceae family [1,2,3]. Consumption of the edible immature inflorescence, also known as the “capitulum” or head, as raw, boiled, steamed, fried, and as an ingredient in a variety of dishes, is the primary commercial use of the plant, as it combines good sensory and health features [4,5]. The high nutritional value of globe artichoke heads is attributed to their low content of lipids and high levels of fibres, including inulin, minerals, vitamins, and bioactive phenolic compounds [6,7,8]. In addition to the edible heads, the cultivation by-products, such as roots, stems, outer bracts, and particularly leaves (representing about 80% of the biomass), can be utilised as raw material for the extraction of valuable compounds for the pharmaceutical and cosmetic sectors [6,9,10]. Furthermore, the artichoke has recently gained interest due to its potential as a biomass crop [11,12].

In 2021, the global production of artichokes was around 1.5 million tonnes, with a cultivated area exceeding 116,000 hectares (ha), of which nearly 60% was in the Mediterranean area [13]. Italy is the largest globe artichoke producer in the world, with an estimated cultivated area of 38,000 ha and a production of roughly 376 kilotonnes (kt), followed by Egypt (17,141 ha/315 kt) and Spain (14,800 ha/214 kt) [13]. Indeed, globe artichoke cultivation is of great importance to the Italian horticultural economy and its area under cultivation is exceeded only by tomato and potato [14]. The regions with the highest production were Sicily (14,230 ha/141 kt), Apulia (11,920 ha/121 kt), Sardinia (6821 ha/38 kt), Lazio (1000 ha/22 kt), and Campania (972 ha/16 kt) [14]. In Italy, approximately 89% of globe artichokes produced are consumed as fresh product, while the remaining 11% is used for processing. Only about 2% of total production is exported, while in recent years there has been a progressive increase in imports, which in 2021 reached approximately 18 kt [14]. Apart from the market needs related to the seasonality of production, the increase in imports is also due to the poor promotion and commercial valorisation of Italian products, including the local productions based on the autochthonous landraces, as well as the inefficient distribution chain.

The most relevant classification criteria for the cultivated artichoke germplasm are harvest time and head shape. The first classifies artichoke genotypes as both autumn and spring (early or “re-bloom” types), or spring exclusively (late or “spring” types). Based on head morphology, four groups can be recognised: “Romanesco,” which has spherical or subspherical, non-spiny heads; “Spinoso,” which has long, sharp spines on both bracts and leaves; “Violetto”, which has violet-coloured heads; and “Catanese”, which has short, elongated, non-spiny heads [15]. The “Spinoso” and “Catanese” varieties include re-blooming genotypes, whereas the “Violetto” and “Romanesco” types are normally harvested in the spring.

Italy has the largest biodiversity of globe artichoke, which has led to the cultivation of several heterogeneous landraces related to the four varietal types, well adapted to the local climatic conditions [16]. Despite this considerable biodiversity, most of the Italian artichoke production is based on a small number of clones/varieties [9,17,18,19].

Increasing market interest in traditional food products is leading to a strong re-evaluation of many traditional crops and neglected genotypes, including many of the globe artichoke landraces [20,21]. Although various efforts have been undertaken to characterise Italian artichoke landraces [9,19,20,22,23,24,25], the majority of them have been inadequately or not at all investigated, so that in many cases, the picture of local biodiversity is unclear. Indeed, landraces are frequently named simply based on the location where they are cultivated [1,19,22] without any study on their genetic variability and structure, or even on their distinction from other local varieties grown in surrounding areas. As a result, landrace names are not always univocal and, in some situations, can result in synonymies or homonymies.

In the Lazio region, globe artichoke production is based mainly on the cultivation of the PGI (Protected Geographic Indication) “CARCIOFO ROMANESCO DEL LAZIO” (Reg. EC n.2006/2002), which is carried out in some predominantly coastal areas in the provinces of Viterbo, Rome, and Latina. The product specification states that the landraces “Castellammare” and “Campagnano”, and their relative clones, must represent the varietal platform of the PGI “CARCIOFO ROMANESCO DEL LAZIO”. Indeed, in the past, the two heterogeneous landraces belonging to the “Romanesco” varietal type were widely cultivated in non-specialized smallholdings of the Lazio region, especially in the area along the Roman coast. Despite the benefits of the PGI certification and the highly desired quality of the heads, these landraces are being rapidly substituted by modern micro-propagated varieties and seed-propagated F1 hybrids that are more productive and mature earlier, in line with market demands [18,19]. In particular, the clone “C3”, selected from the “Castellammare” population for its earliness, has replaced the traditional “Romanesco” landraces in most of the artichoke plantations in Lazio region [19,24,26]. This has resulted in a substantial loss of local genetic resources, with the possibility of an increased susceptibility of the cultivated material to biotic and abiotic stresses. Indeed, the “Romanesco” germplasm exhibits high levels of intra-landrace genetic variability [19,24,27]. Moreover, several of these landraces are resistant to endemic diseases, such as *Verticillium dahliae* Kleb. and Artichoke latent Virus (ArLV) [18,24]. This emphasizes the need to preserve landraces because their high variability enables them to respond to abiotic and biotic challenges and adapt to low-input agricultural systems.

In addition to “Castellammare” and “Campagnano”, another landrace, known as “Carciofo Ortano”, is still cultivated in the Lazio region. The name derives from Orte, a little village located in the province of Viterbo (Central Italy), the place where it was found and where it is still grown, especially in the plains along the riverbanks of the Tiber River. The origin of the “Carciofo Ortano” is unknown, but from some sources from the late nineteenth century [28], we know that in this area there was an intensive and well-established production of globe artichokes, with an estimated cultivation area of more than 30 ha. Despite the massive abandonment of the agricultural sector since the 1960s, the “Carciofo Ortano” has survived thanks to the efforts of a few small farmers and horticulturists, who have continued to cultivate, mainly for family use, this landrace inherited from their parents and grandparents. However, its survival is strongly threatened due to several factors, including the advancing age of the keepers, the small area occupied by artichoke fields, mostly located in local vegetable gardens or small family farms, and the introduction of the commercial varieties.

With a view to the recovery, protection and valorisation of “Carciofo Ortano”, the aims of the present study were: (i) to survey the farms and family gardens where the landrace is still grown; (ii) to develop a molecular methodology using SSR and ISSR markers for its distinctiveness with respect to the cultivars or landraces grown in Italy; (iii) to evaluate the genetic diversity and structure of the landrace in order to accurately define the most suitable strategies to be adopted for its conservation “in situ”; and (iv) to identify the representative genotypes of the genetic variability found in the landrace to be reproduced “in situ” by setting up a field gene bank. The field gene bank was established with two main objectives: to allow the conservation of the genetic diversity of the “Carciofo Ortano” and to establish the morphological and nutritional characterisation of the landrace by using material grown in the same experimental field.

## 2. Materials and Methods

### 2.1. Identification and Survey of Smallholdings and Family Gardens Growing the Landrace “Carciofo Ortano”

Thanks to both the support of the local farmers’ association “Horti in Tiberi” and of the technicians of ARSIAL (Lazio Region Agency for Agricultural Development and Innovation), 20 smallholdings and family gardens located in the countryside of Orte municipality were identified where, according to the farmers, the landrace “Carciofo Ortano” is still grown. All the identified collection sites were georeferenced with GPS (Global Positioning System) and the data were collected into a GIS (Geographic Information System), allowing the design of a map of smallholdings and family gardens where “Carciofo Ortano” is currently cultivated (Figure 1). During the field visits to collect plant material for genetic/molecular analysis (see below), farmers were interviewed to gather knowledge related to history of cultivation of globe artichoke in that area, which is still passed down orally. Moreover, the interview aimed to collect information regarding the cultivation time of the landrace, any knowledge about the source of the original propagation material, traditional agronomical practices, and use of harvested main and secondary heads.

### 2.2. Molecular Characterisation of the “Carciofo Ortano” Landrace

#### 2.2.1. Plant Material

A total of 73 plants were selected in the artichoke plantings located in the 20 smallholdings and family gardens (Appendix A). The number of plants, which was randomly selected in each artichoke planting, varied according to its size, with higher numbers (6 to 13 plants) used for plantings of 1000–1500 m^2^ and lower numbers (1–5 plants) used for smaller plantings (50–500 m^2^) (Appendix A). In April and May of 2020, the selected plants were photographed and marked with a stake bearing a tag reporting the plant number and the name of the keeper farmer (Appendix A). Young leaves were collected from the 73 chosen plants, immediately frozen in liquid nitrogen, and stored at −80 °C until DNA extraction. In the molecular analysis, 17 accessions of landraces/clones belonging to the four varietal types cultivated in Italy were also included as reference genotypes: “Romanesco”, “Violetto”, “Spinoso”, and “Catanese” (Appendix A) [29,30,31,32,33,34]. Except for the “Castellammare Sezze” accession, which was collected from a farm in Sezze, in the province of Latina (Lazio region), all the remaining 16 accessions were collected from the experimental field of ARSIAL located in Cerveteri, in the province of Rome. The largest number of accessions of landraces/clones used as references belonged to the “Romanesco” type, since preliminary morphological surveys indicated that the “Carciofo Ortano” could be assigned to this varietal typology. Besides three accessions of “Campagnano” and “Castellammare”, and several clones selected from the two landraces, which represent the varietal platform of the PGI “CARCIOFO ROMANESCO DEL LAZIO”, we included in the analysis an accession of the landrace “Montelupone”, cultivated in the homonymous municipality in the province of Macerata (Marche, Central Italy) (Appendix A). Young leaves from the accessions used as reference genotypes were collected and stored as described above.

#### 2.2.2. DNA Extraction and Molecular Markers (SSR and ISSR) Analysis

Total genomic DNA was isolated from 200 mg of frozen tissue, using the NucleoSpin^®^ Plant II kit (Macherey-Nagel, Düren, Germany) following the manufacturer’s instructions. To assess the integrity and concentration of DNA, a 0.8 (*w*/*v*) agarose gel stained with ethidium bromide (0.001%) and a Nanodrop Bioanalyzer ND1000 (ThermoScientific, Waltham, MA, USA) were used. All DNA samples were kept at −20 °C until usage.

Twelve SSR loci were selected from previous genetic studies of globe artichoke germplasm collections [17,19,35,36,37,38,39] based on their high values of PIC (Polymorphic Information Content) (Appendix A). SSR primer pairs were multiplexed, labelling their forward primer with FAM, TAMRA, and JOE fluorescent dyes (Eurofins Genomics, Ebersberg, Germany). PCR reactions were performed as described previously [40], using the specific annealing temperature of each primer pair (Appendix A). Amplified SSR products were genotyped using an ABI PRISM 3500 sequencer (Applied Biosystems, Waltham, MA, USA) and sized in accordance with GeneScan LI500Liz standard using Gene Mapper 4.0 software (Applied Biosystems).

For the ISSR markers, DNA samples were amplified with ten primers (Appendix A), two of which were previously employed to study the genetic variability in different genotypes of the genus *Platanus* [41], and eight from the British Columbia University (UBC) collection, which had been used to genetically characterise different artichoke germplasm collections [9,17,19,24]. PCR reactions and the analysis of amplification products were performed according to Alicandri et al. [40].

### 2.3. Planting of Collection Experimental Field for Morphological and Nutritional Analyses

Based on molecular analyses, 32 plants from 11 smallholdings/family gardens were selected as representative genotypes of the diversity found within the “Carciofo Ortano” landrace. In November of 2020, five to seven offshoots (named “carducci”) were taken from each of the 32 selected plants and, according to local practices, the distal part of leaves were removed. Offshoots were then placed into boxes, covered with damp cloths, and brought to the site of the collection experimental field located in the countryside of the Orte municipality (42.47446 N, 12.33293 E), at an altitude of about 130 m above sea level (Figure 1). There, they were transplanted together with offshoots collected from plants grown at the experimental field of ARSIAL located in Cerveteri of the four landraces/clones included in the varietal platform of the PGI “CARCIOFO ROMANESCO DEL LAZIO” used as reference genotypes: “Campagnano”, “Castellammare”, “C3”, and “Grato 1”. Offshoots were planted in rows spaced 1.00 m apart with row spacing of 2.5 m, to allow the passage of tractors for mechanical works. The material was arranged in a randomised block design with three replications, with each experimental unit consisting of five plants. Field experiments were conducted under low resource inputs (two irrigations/year in April and August consisting of 60 mm of water each, organic fertilisation with 50 kg of N ha^−1^, and without herbicides, pesticides, and gibberellic acid), according to the local agronomic practices.

### 2.4. Morphological Analysis

Six plants for six different genetic groups (genotypes) identified by molecular analysis in the “Carciofo Ortano” landrace (two plants for each replicate) and six plants for each of the four landraces/clones used as reference genotypes (two plants per replicate) were randomly selected in each plot of the collection experimental field and assessed agro-morphologically. Each plant was phenotyped using standard UPOV descriptors and a well-defined group of complementary “Romanesco” type descriptors according to previous studies [9,19,24] (Appendix A). For each plant, morpho-physiological data were recorded once or twice a week during the period between the time of appearance of the central flower head and the commercial maturity of the tertiary heads (from April to May) in the 2022 growing season. Earliness (main head date of appearance and maturity) was established counting the number of days from a reference date, which was chosen arbitrarily as 1 January to the main flower head appearance and harvesting.

### 2.5. Nutritional and Chemical Analysis of Primary Flower Heads

#### 2.5.1. Sampling and Preparation of Plant Material

Three primary flower heads from twelve plants assigned to four distinct genetic groups of “Carciofo Ortano” (three plants for each replicate) and twelve from the plants of the four reference genotypes (three flower heads per replicate) were harvested during the commercial maturation stage of each genotype in the 2022 growing season. After the removing of the external nonedible bracts and the cutting of stems to one cm, the flower heads were washed with distilled water and cut transversely into four parts. After their weighing, all the samples were then freeze-dried and ground to approximately 0.5 mm diameter fine powder, except for the fresh plant material used for determination of the moisture content. The freeze-dried and ground samples were transferred into 50 mL polypropylene tubes, covered with silver paper, and stored at −80 °C until use.

#### 2.5.2. Nutritional Value and Mineral Composition Analysis

The basic chemical composition of the flower heads, including moisture, ash, lipid, protein, dietary fibre, and carbohydrate contents, was determined mainly according to procedures established by the Association of Official Analytical Chemists [42]. Approximately 10 g of fresh plant material was dried in a forced-air oven at 70 °C for three days until a constant weight was reached to determine the moisture content. According to the AOAC Official Method 942.05, 2 g of freeze-dried and ground sample weighed in a porcelain crucible was put in a muffle furnace at 600 + 2 °C for 2 h to determine ash content. The samples were then allowed to cool at ambient temperature in a desiccator before being weighed to calculate the ash content. For the total lipid content, samples (5 g) were extracted with CHCl_3_ using the Soxhlet apparatus for 4 h and then dried using a rotary evaporator to obtain the lipid extracts, which were then weighed to determine the amount of extracted fat [21]. Nitrogen concentration was determined using the Kjeldahl method following the AOAC Official Method 920.54, and total protein content was calculated using a nitrogen factor of 6.25. Insoluble and soluble high-molecular-weight dietary fibre (IHMWDF and SHMWDF) were determined using enzymatic hydrolysis (Megazyme kit, K-TDRF, Megazyme, Wicklow, Ireland) in accordance with the official AOAC method 991.43 [43]. The sum of IHMWDF and SHMWDF was used to compute total high-molecular-weight dietary fibre (THMWDF). Finally, total carbohydrates were obtained by subtracting moisture, ash, lipid, protein, and dietary fibre contents to 100 [44].

The total inulin content was determined following the method of Steegmans et al. [45], modified according to Pandino et al. [4]. The method is based on inulin/oligofructonase enzymatic hydrolysis with fructanase, followed by spectrophotometric determination according to the Megazyme protocol. In particular, 1 g of freeze-dried sample was combined with 40 mL of boiling water, the pH was set to 7.0 with 50 mM of KOH, and the solution was maintained at 85 ± 2 °C for 15 min. After cooling at room temperature, the volume was increased to 100 mL with deionized water. An aliquot was incubated for 30 min at 40 ± 2 °C with sucrase, and another for 60 min at 60 ± 2 °C with fructanase, and in both cases the absorbance was measured at 340 nm. The total inulin content was calculated according to Steegmans et al. [45].

All the samples were analysed in duplicate, and their nutritional components expressed as g 100 g^−1^ of fresh weight (FW).

For mineral content, each sample (500 mg) was digested with 2 mL of 30% (m/m) H_2_O_2_, 0.5 mL of 37% HCl, and 7.5 mL of HNO_3_ 69% solution. The acid digestion was carried out using a high-pressure laboratory microwave oven Mars plus (CEM srl, Cologno al Serio, Italy). The temperature was linearly increased from 25 to 180 °C over 37 min before being maintained at 180 °C for 15 min. The digested samples were diluted with ultrapure water to a final volume of 25 mL. For mineral quantification, an ICP-OES (8000 DV, PerkinElmer, Shelton, CT, USA) with an axially viewed configuration and ultrasonic nebulizer was utilised. Multi-elemental, high-purity grade was obtained from CaPurAn (CPAchem Ltd., Bogomilovo, Bulgaria) and the external calibration solutions were prepared from standard certified elemental solutions (CaPurAn). All the samples were analysed in duplicate, and the data expressed as mg 100 g^−1^ FW.2.5.3. Total Polyphenols and Flavonoids Content

About 2 g of the powdered freeze-dried samples were extracted with 60 mL of 100% methanol using ultrasound-assisted extraction (UAE) (200 W) three times (30 min each time), followed by filtration through a 0.45 µm Whatman filter paper to remove all the solids from the extracts. The samples were stored at −20 °C until use.

Total phenolic content (TPC) was determined using the Folin–Ciocalteu colorimetric method [46] with minor modifications. Briefly, a 20 µL aliquot of each methanol extract (with a range concentration from 0.01 to 5 mg of extract dry weight per mL) was mixed with 100 µL of the Folin–Ciocalteu reagent and the flasks were shaken vigorously. After 8 min, 80 µL of 7.5% Na_2_CO_3_ was added, and the mixture was vortexed. After reacting for 2 h at room temperature (20 ± 2 °C) in the dark, absorbance was measured at 765 nm using an Ultrospec 1000 spectrophotometer (Phamacia Biotech, Uppsala, Sweden). All the tests were performed in triplicate. The TPC contents were expressed as mg gallic acid equivalents (GAE) per g^−1^ of dry weight (DW).

For the determination of total flavonoid content (TFC), the aluminium chloride (AlCl_3_) colorimetric method was used [30]. In brief, an aliquot of 500 µL methanolic extract for each sample (with a range concentration from 0.01 to 5 mg of extract dry weight per mL) were mixed with 1.5 mL of methanol, 2.8 mL of water, 100 µL of potassium acetate (1 M), and 100 µL of aluminium chloride (10% in methanol). After reacting for 30 min at room temperature (20 ± 2 °C) in the dark, absorbance was measured at 415 nm using an Ultrospec 1000 spectrophotometer (Phamacia Biotech, Uppsala, Sweden). All the tests were performed in triplicate. The TFC content was expressed as mg of rutin equivalents (RUE) per g^−1^ DW.

### 2.6. Data Analysis

#### 2.6.1. Molecular Data

For SSR codominant markers, each allele was reported according to its size (pb) and their screening power was evaluated by determining the PIC index [47], using Power Marker 3.25 software [48]. Genetic diversity per locus was evaluated by the following parameters: number of observed alleles per locus (N_a_), number of rare (allele frequency < 0.05) and private alleles, major allele frequency (MAF), and expected and observed heterozygosity (H_e_ and H_o_), by using Power Marker 3.25 [48] and GenAlEx6 [49].

For each of the ISSR markers the DNA fragments (bands) in all genotypes were scored as present (1) or absent (0), and the raw data were imported into a Microsoft EXCEL spreadsheet to build a binary matrix. The genetic diversity for each ISSR locus was assessed using the following parameters: Number of Total Bands (NTB) obtained for each ISSR primer, Number of Polymorphic Bands (NPB), Percentage of Polymorphism (% Pol), number of rare (frequency < 0.05) and private bands (alleles), MAF, He, and PIC, with these latter three parameters determined using Power Marker 3.25 [48].

For the combined analysis of SSR and ISSR markers, a single matrix was created by converting the size of the amplification fragments of the SSR markers into 1/0 values (presence/absence). The genetic distances for phylogenetic relationships among the different genotypes were estimated using the coefficient of Nei [50]. The obtained distances matrix was used to construct a phylogenetic tree using the UPGMA (unweighted pair-group method with arithmetic averages) clustering method in MEGAX software [51]. The reliability of the tree topology was assessed via bootstrapping over 1000 replicates using the PAUP* 4.0 software [52].

To evaluate the genetic structure of the landrace “Carciofo Ortano”, a Bayesian-based clustering method was applied on multi-locus SSR and ISSR data using STRUCTURE v. 2.3.4 software [53], as described by Ciaffi et al. [54]. The most likely number of clusters (K) was determined using the procedure of Evanno et al. [55], which proposed an ad hoc statistic, ΔK, to reduce a possible overestimation of subgroup by STRUCTURE. Samples with membership probabilities ≥0.75 were assigned to the corresponding subgroup.

Genetic diversity of “Carciofo Ortano” genotypes was analysed by classifying the genotypes into two groups, according to the results of model-based cluster analysis of SSR and ISSR loci. The genetic diversity of each of the two groups, named Orte 1 and Orte 2, was assessed by calculating the effective number of alleles (N_e_), the number of private alleles, the Shannon’s Information Index (I), and gene diversity (H_e_) for both the ISSR and SSR markers in GeneAlEx6 [49] and Power Marker 3.25 [48]. In addition, the observed heterozygosity (H_o_) was also calculated for SSR markers in GeneAlEx6 [49]. To test the significance of the differences in the genetic parameters between the two groups, the Kruskal–Wallis non-parametric test was performed by using JMP PRO 15 (©SAS Institute Inc., Cary, NC, USA). The analysis of molecular variance (AMOVA) using GeneAlEx6 [49] was also performed to partition the genetic variation between and within the two groups of genotypes. The variance components were tested statistically by non-parametric randomisation tests using 9999 permutations.

#### 2.6.2. Morphological Data

Morphological data were analysed using one-way ANOVA and principal component analysis (PCA). The means were compared by using Tukey’s pairwise tests at a significance level of *p* ≤ 0.05. Shapiro–Wilk test was used to assess the normality of distribution of the observations. The coefficient of variation (i.e., the value of the standard deviation of the mean divided by the mean), expressed as percentage, was used to analyse the variability of the morphological quantitative traits between the analysed genotypes. All these statistical analyses were performed using JMP PRO 15 (©SAS Institute Inc., Cary, NC, USA).

#### 2.6.3. Nutritional and Chemical Data

Data of nutritional and chemical analyses were submitted to one-way ANOVA by using JMP PRO 15 (©SAS Institute Inc., Cary, NC, USA), and the means were compared by using Tukey’s pairwise tests at a significance level of *p* ≤ 0.05. The coefficient of variation was used to analyse the variability of nutritional and chemical values between the analysed genotypes.

## 3. Results and Discussion

### 3.1. Genetic Analysis by SSR and ISSR Markers

#### 3.1.1. Genetic Diversity

Among the twelve selected SSR loci, two (CMAFLP_18 and Cmal_07) were monomorphic and one (CLIB_02) showed amplification profiles that are difficult to interpret; thus, Table 1 reports the main genetic parameters of the nine remaining polymorphic loci.

In the 90 genotypes analysed, the identified alleles for the codominant SSR markers were 65 in total, and the number of alleles ranged from 4 (Cmal_06 and Cmal_08) to 10 (CELMS_40), with an average of 7.22 alleles per locus (Table 1). Approximately 31% (20/65) of alleles were rare, with a frequency lower than 5% (Appendix A); as a result, the major alleles had an overall average frequency of 62% (Table 1). Five of the SSR rare alleles were specific only for the landrace “Bianco Ostuni” (Appendix A). The expected heterozygosity (H_e_) ranged from 0.165 (CELMS_08) to 0.658 (CDAT_04), and 6 loci (CDAT_02, CDAT_04, CELMS_05, CDAT_01, CELMS_40 and CsPal_03) showed higher values than the mean value of 0.524 (Table 1).

It is worth nothing that the total number of alleles, and the average N_a_ and H_e_ per locus detected here, were higher than the corresponding values reported by De Felice et al. [56] and Crinò and Pagnotta [19], who analysed 28 accessions of cultivars/clones belonging to the 4 varietal types cultivated in Italy and 10 clones belonging to the “Romanesco” typology, respectively. Furthermore, mean N_a_ and H_e_ across the nine SSR loci were higher compared to those observed in other studies carried out on large globe artichoke worldwide collections [17,57], highlighting the high level of genetic diversity present in our collection, and in particular in the “Carciofo Ortano” landrace.

The observed heterozygosity (H_o_) ranged from 37% for CDAT_02 to 99% for CDAT_01, with an average of 82% (Table 1). The high level of heterozygosity detected is comparable with the values found in several studies carried out on Italian as well as worldwide globe artichoke collections [17,19,56,57]. These results are expected since globe artichoke is a highly outcrossed species, which is mainly propagated vegetatively by using the “carducci” (basal shoots) or the “ovoli” (semi-dormant shoots with a limited root system).

The SSR markers differed in their ability to detect genetic variation among the 90 genotypes analysed, as shown by the differences in the values of N_a_, H_e_, MAF, and PIC (Table 1). Overall, the most informative SSR loci were CDAT_04 and CELMS_40, which showed the highest PIC (0.609) and H_e_ (0.658 and 0.650, respectively) values, and among the lowest MAF values (Table 1). Importantly, these loci are among those with the highest number of alleles: 10 (CELMS_40) and 8 (CDAT_04) (Table 1). Based on the calculated genetic parameters, CDAT_01, CELMS_05, CDAT_04, and CsPal_03 can also be considered good markers with the PIC values comprised between 0.563 and 0.594 (Table 1).

Among the ten selected ISSR primers, four of these (UBC_823, UBC_837, UBC_836, and UBC_853) were not considered further as they showed complex or otherwise unclear amplification patterns, which were difficult to interpret in many situations. Therefore, statistical analyses were carried out for the six ISSR primers that showed clear and reproducible amplification patterns and for which the main genetic parameters are shown in Table 1. As an example, Appendix A shows the polymorphic patterns of three out of six ISSR primers from different “Carciofo Ortano” genotypes and landraces/clones used as references. Based on the ISSR analysis, 76 bands, ranging from 300 bp to 2 Kb in size, were generated across the 90 genotypes analysed, with an average of 12.67 per primer (Table 1). The number of bands varied from 10 (primer UBC_848) to 17 (primer ISSR_24). ISSR_28 and ISSR_24 primers produced the lowest (7) and the highest (17) number of polymorphic bands, respectively (Table 1). Overall, 72 polymorphic bands were detected (94% polymorphism). Approximately 32% (24/76) of bands (alleles) were rare, having a frequency lower than 5%, with five specific to four different genotypes (Appendix A).

H_e_ and PIC values for the dominant ISSR markers had relatively low and uniform values, with an average of 0.242 and 0.196, respectively (Table 1). In particular, H_e_ values ranged from 0.040 (ISSR_28) to 0.363 (UBC_848), whereas PIC values were ranged between 0.039 (ISSR_28) and 0.282 (UBC_848). Despite the high number of amplicons detected by the dominant ISSR markers, and therefore the possibility to explore a wider genomic region, they were less informative (in terms of PIC and H_e_ values) than the co-dominant SSR markers due to their uncertain allelic phase [17].

#### 3.1.2. Genetic Relationship among Genotypes

The UPGMA dendrogram based on the genetic distances between the 90 genotypes, estimated by Nei’s coefficient (1973) using both SSR and ISSR molecular markers, is reported in Figure 2. The dendrogram showed 2 main clusters (CL I and CL II), which included 65 of the 73 plants sampled in the Orte municipality and all the landraces/clones belonging to the “Romanesco” type (Figure 2). These two main clusters were clearly distinguishable from the accessions used as control genotypes belonging to the “Violetto” (“Tardivo di Pesaro”), “Catanese” (“Catanese” and “Brindisino”), and “Spinoso” (“Spinoso Sardo”) types and the two accessions for which the varietal type was not defined, “Bianco di Ostuni” and “Carciofino di Pontecorvo”, the latter being closely associated with “Spinoso Sardo” (Figure 2). Of the eight plants sampled in the municipality of Orte that were not assigned to the two main clusters, one (F6 P2) showed a remarkable genetic similarity with “Tardivo di Pesaro”, while the remaining seven formed two distinct groups, the first including F18 P1–4, and the second including F1 P2 and F6 P1–2, both clearly separated from all the reference genotypes (Figure 2). These results suggested that these eight plants could not be considered to belong to the “Carciofo Ortano” landrace.

Based on the Nei’s genetic distances, the large cluster CL I containing 38 of the 73 plants sampled in the Orte countryside could be further divided into 3 sub-clusters, indicated as IA, IB, and IC.

In addition to a plant sampled in Orte (F1 P1), the first sub-cluster (CL IA) included the variety “Terom” (varietal type “Violetto”) and the two clones “Grato 1” and “Leonardo” (varietal type “Romanesco”). These apparently contradictory results can be explained by considering the origin of the three control genotypes belonging to two different varietal types. “Grato 1” is a clone selected from plants obtained by cross-pollination of clones belonging to the “Castellammare”, “Campagnano” and “Violetto di Toscana” landraces, the latter being used as mother plant [29]. “Leonardo” is a clone selected from “Grato 1”, while “Terom” is a clone obtained by selection of a progeny from seeds collected from “Violetto di Toscana” landrace [29]. Based on these results, we can assume that the F1 P1 plant could be genetically related to the “Violetto di Toscana” landrace.

In addition to the “Campagnano” and “Castellammare” landraces and the clones derived from them (“C3”, “Raffaello”, “Michelangelo”, and “Donatello”), which constitute the varietal platform for the cultivation of PGI “CARCIOFO ROMANESCO DEL LAZIO”, the second sub-cluster, CL IB, included 34 plants sampled in the municipality of Orte. In turn, this sub-cluster could be divided into two distinct genetic groups, generally related to the provenance of the genetic material. The first comprised the seven accessions of landraces/clones of the PGI “CARCIOFO ROMANESCO DEL LAZIO” and six plants collected from three different farms in Orte countryside (F20 P1–4, F12 P1, and F13 P2). The second group of the sub-cluster CL IB, on the other hand, included 28 plants all sampled in the Municipality of Orte, mainly from the farms 3 (P1–3), 10 (P1–3), 17 (P1–6), 19 (P1–4), and 4 (P1–6 and P8–13) (Figure 2).

Finally, the sub-cluster CL IC contained three plants from three different Orte farms (F4 P7, F12 P2 and F8 P2) (Figure 2).

The cluster CL II comprised 27 plants sampled in the municipality of Orte that were found to be genetically related to the “Montelupone” landrace also belonging to the “Romanesco” type (Figure 2). The cluster CL II in turn could be divided into two sub-clusters referred to as IIA and IIB. The first sub-cluster contained, in addition to “Montelupone”, the only plant sampled in the farm 4 (F4 P1), one of the two plants from the farm 13 (F13 P1) and four of the five plants from the farm 8 (F8 P1 and F8 P3–5). The second sub-cluster, CL IIB, comprises 21 plants from 6 different farms (F5 P1–8, F16 P1–2, F15 P1–3, F14 P1–2, F7 P1–5, F11 P1) (Figure 2).

#### 3.1.3. Genetic Structure of the “Carciofo Ortano” Landrace

A Bayesian-based clustering method was applied to multi-locus SSR and ISSR data using the STRUCTURE program to infer the genetic structure of the “Carciofo Ortano” landrace by defining the numbers of genetic groups in the dataset, assign the individuals to each of the identified groups, and identify admixture individuals. Only the SSR and ISSR data of the 64 genotypes sampled in the Orte municipality assigned, based on the UPGMA analysis, to the 2 main clusters CL I (sub-clusters IB and IC) and CL II and those from the 8 landraces/clones of the “Romanesco” type closely associated with them (Figure 2), were used in this analysis.

Bayesian analysis revealed that the 72 genotypes could be represented by 2 (K = 2) genetic groups (Appendix A), according to the 2 major clades obtained by the clustering analysis based on the UPGMA method (Figure 3).

The first group (labelled green) included 37 plants sampled from Orte farms and the 7 accessions of landraces/clones of the PGI “CARCIOFO ROMANESCO DEL LAZIO” assigned to the main cluster CL I, while the second (labelled red) comprised the “Montelupone” landrace and 27 plants from Orte farms assigned to the main cluster CL II (Figure 3). All genotypes were uniquely assigned to one of the two groups based on the membership probability Q value, which was in all cases >0.75 for individuals assigned to each of the two genetic groups (Appendix A). These results clearly indicated the complex genetic structure of the “Carciofo Ortano” landrace, in which two different genetic populations can be distinguished. The first, named Orte 1, is genetically closer to the “Campagnano” and “Castellammare” landraces and the clones derived from them, which form the varietal basis for the cultivation of the PGI CARCIOFO ROMANESCO DEL LAZIO, and the other, called Orte 2, is closer to the “Montelupone” landrace, cultivated in the district of the homonymous municipality in the province of Macerata (Marche, Italy).

The diversity analysis of the two populations by considering the SSR markers revealed that the Orte 2 group showed significantly higher values for the effective number of alleles (N_e_), Shannon index (I), and expected heterozygosity (H_e_) than the Orte 1 group, while no significant difference was detected for the observed heterozygosity (H_o_) among the two groups (Table 2).

Of the 27 total alleles detected across the 9 SSR loci in the 64 genotypes, 15 were specific to 1 of the 2 populations, with the Orte 2 population showing the highest number of private alleles (11) (Table 2 and Appendix A). For the ISSR-dominant markers, the Orte 1 population showed higher values of N_e_, I, and H_e_ than Orte 2, but the differences were not statistically significant following the Kruskal–Wallis test (Table 2). In contrast to the SSR markers, the 37 genotypes of the Orte 1 population revealed a higher number of ISSR private alleles (15) than the 27 genotypes of the Orte 2 population (6) (Table 2 and Appendix A). Interestingly, some of the SSR and ISSR private alleles of the two populations were specific to a single genotype or common to a small group of individuals belonging to the same genetic group (Appendix A and Appendix A). These SSR and ISSR alleles can be very useful for future programs of clonal selection within the two populations, enabling, for instance, the development of simple PCR-based assays for clone identification.

AMOVA analysis showed different results for the two typologies of molecular markers. In the case of the co-dominant SSR markers, the greatest part of the variability (80.55%) was attributable to differences among populations, while in the case of the dominant ISSR markers, only 34.94% of the total variation was among populations, with the largest part (about 65%) attributable to differences within them (Table 3). These results clearly showed that the SSR markers mainly allowed the distinction of genotypes between the two populations, while the ISSR markers provided a better resolution of genetic variability within the two groups, indicating the specific usefulness and complementarity of the two types of markers used.

### 3.2. Setting up of the Collection Experimental Field

Based on Nei’s genetic distances, the 37 and 27 genotypes assigned to the 2 main clusters, CLI and CLII (Orte 1 and Orte 2 populations, respectively), could be further subdivided into 14 distinct genetic groups indicated as GGR 1–14 (Figure 3A). These genetic groups included plants of the two populations of the “Carciofo Ortano” landrace that were genetically identical or at least very similar to each other (Figure 3A). Four plants (F8 P2, F13 P1, F2 P1, and F11 P1) could not be assigned to the previous genetic groups and were therefore indicated with the abbreviation Sep 1–4 (genetically separated) (Figure 3A). Some of the genetically similar/identical plants belonging to the 14 genetic groups were used for the sampling of offshoots (“carducci”), which were planted in November 2020 for the constitution of a field gene bank. Specifically, plants belonging to 9 of the 14 identified genetic groups were selected, from which enough offshoots (at least 15 for each genetic group) were taken for the 3 replications in the experimental design of the collection field. These plants were as follows: F17 P1–6 (GGR4 and GGR7), F4 P8–12 (GGR5), F19 P1–4 (GGR6), F3 P1–3 (GGR 8), F8 P1 and P3–4 (GGR 10), F7 P2–4 (GGR11), F7 P1/P5, F15 P1/P3 (GGR12), and F5 P1–4 (GGR14). The field gene bank also included four landraces/clones belonging to the “Romanesco” type used as reference genotypes: “Campagnano”, “Castellammare”, “C3”, and “Grato 1”.

### 3.3. Morphological Characterisation

Two plants for each of the three replicates for six different genetic groups, three belonging to the Orte 1 population (GGR4, GGR5, GGR8) and the others to the Orte 2 population (GGR10, GGR11 and GGR12), and six plants for each of the four landraces/clones used as reference genotypes (two plants per replicate), were randomly selected for the morphological characterisation.

Ten out of the twenty-five qualitative morphological descriptors measured allowed differences to be detected among the ten genotypes analysed, with only five able to discriminate the two populations of the “Carciofo Ortano” landrace (Appendix A). In particular, Orte 1 genotypes showed spherical-shaped heads with a rounded apex in three of the four landraces/clones used as reference genotypes (“Campagnano”, “Castellammare”, and “C3”), while Orte 2 genotypes had elliptical heads with a flattened apex (Figure 4).

These differences were confirmed by the head shape index (head length/diameter ratio) determined for the 10 genotypes (see below). In addition, the two populations differed in terms of the shape of the outer bracts, which were as long as broad in the Orte 1 genotypes and broader than long in Orte 2 genotypes, and for the shape of the receptacle, which was slightly depressed in Orte 1 genotypes and strongly depressed in the Orte 2 genotypes (Appendix A). No differences were found in leaf morphological characteristics among the ten genotypes analysed (Appendix A). Indeed, it is not surprising that traits such as spines, colour, and shape of the leaves did not vary among genotypes of the “Romanesco” typology, including the two populations of the “Carciofo Ortano”.

Regarding the quantitative morphological traits, a pair-wise similarity matrix among the 10 genotypes considered was generated and a PCA was subsequently applied. The first three principal components (PCs) gave eigenvalues higher than one, and together accounted for about 89% of the total variance (Appendix A). The first component (51.72% of variance) included contributions from the weight and the height of central and primary heads (WCH, LCH, WH1, and LH1), and the total production (Y). The second component explained 23.66% of the variance and was correlated with the shape index of both central and primary heads (LD ratio and LD1 ratio), the diameter of receptacle (RecD), and the main stem diameter (MSD). The third component (13.86% of the variance) entailed the time of central head appearance and head maturity (THA and THM), the plant height (PH), the main floral stem height (MSL), and the distance between the first fully developed leaf and the central head (DHL). The distribution of the genotypes against the first three discriminant functions showed that three groups could be identified (Figure 5), with the Orte 2 genotypes located on the upper left, those from the Orte 1 population on the upper right, and the four landraces/clones used as reference genotypes on the lower right (Figure 5).

Table 4 shows the results of ANOVA analysis with the statistically significant differences among genotypes for the 12 morphological quantitative traits that, according to the PCA, showed the highest correlation with the first 3 PCs, while the results for all 18 quantitative traits are reported in Appendix A.

The Orte 1 and Orte 2 genotypes showed the highest and the lowest values for plant height (PH), respectively. “Castellammare” and “C3” revealed the biggest main floral stem diameter (MSD), while no significant differences among Orte 1 and Orte 2 genotypes were found for this trait. Orte 2 genotypes, together with “Castellammare” and “C3”, exhibited the lowest distance between the youngest leaf on the main floral stem and the central flower head (DHL). Regarding the shape of both central and primary heads (LD and LD1 ratios), “Grato 1” was the genotype with the most elongated heads. In contrast, Orte 1 genotypes, together with “Campagnano”, “Castellammare”, and “C3”, showed round-shaped heads, whereas Orte 2 genotypes had the most elliptical heads, confirming the initial visual examination of flower capitula. With reference to yield, Orte 1 genotypes, together with “C3”, showed the highest values of total head weight per plant (Y), as a result of the higher weight of central and primary heads (WCH and WH1). By contrast, Orte 2 genotypes exhibited the lowest values for the total production (Y) and the weight of central and primary heads (WCH and WH1). Finally, as for earliness (THA and THM), “C3” and “Castellammare” were the earliest genotypes, followed by the Orte 1 genotypes, “Campagnano”, and “Grato 1”, whereas the Orte 2 population included the latest genotypes.

In accordance with the molecular analyses, these findings confirmed the presence of two different populations within the “Carciofo Ortano” landrace, which, in turn, can be phenotypically distinguished from the four main landraces/clones included in the varietal platform for the cultivation of PGI “CARCIOFO ROMANESCO DEL LAZIO”. Indeed, as shown by PCA and ANOVA results, particular morphological quantitative traits, such as plant height, head shape index, head yield, and earliness, although potentially affected by the environment conditions, appeared to be useful when differentiating the genotypes belonging to the “Romanesco” typology as observed in other studies [19,24].

### 3.4. Nutritional and Chemical Characterisation

Nutritional values of the heads of the studied artichoke genotypes are reported in Table 5. Moisture content, which could influence texture and the overall quality of the commercial heads [7], differed significantly among the analysed genotypes, ranging from 80.25 to 83.38%, with Orte 1 and Orte 2 genotypes showing the highest and the lowest values, respectively. Petropolous et al. [7] reported similar moisture content values of artichoke heads in the range of 79–84% for various landraces and modern varieties cultivated in Greece. In contrast, Bonasia et al. [18] and Pandino et al. [4], by analysing new seed-propagated hybrids and clones selected from a population of “Violetto di Sicilia”, respectively, reported higher values (84–90%) compared to those obtained in this study. Moisture and therefore dry matter content of artichoke heads may be affected by the head morphology, in which denser outer bracts and compactness of head may minimise water loss [58], as well as growing conditions, including irrigation status and fertilizer applications [59]. Ash content also varied between genotypes, with “Campagnano” and Orte 2 genotypes showing the highest and the lowest values, respectively, whereas no significant differences among the studied genotypes were found for fat content. Protein content ranged from 2.96 to 3.25 g 100 g^−1^ FW, in substantial accordance with results found in the literature [4,7,21]. Additionally, the carbohydrate content of heads from the genotypes differed significantly, ranging from 6.17 to 8.20 g 100 g^−1^ FW, with Orte 2 genotypes exhibiting significantly higher values than Orte 1 genotypes and the landraces/clones used as controls. Comparable values of total carbohydrates have been reported for the landrace “Capuanella” belonging to the “Romanesco” typology [21], whereas significantly higher values in the range of 11–19 g 100 g^−1^ FW were found for heads of different genotypes included in the previously mentioned artichoke germplasm collection from Greece [7]. These discrepancies could be attributed to the diverse genotypes analysed and to the different methods used to calculate the total carbohydrates. Indeed, according to Dosi et al. [21], in our study, carbohydrate content was obtained by subtracting, in addition to moisture, ash, lipid, and protein, the total high-molecular-weight dietary fibre (THMWDF).

It is now widely acknowledged that dietary fibre (DF) is an essential nutrient for humans, due to its association with health benefits such as weight control, satiety, stabilisation of blood glucose levels, reduction of cholesterol levels, protection from some types of colon cancer, and prebiotic activity [60]. According to the CODEX Alimentarius Commission [61,62], DF constituents can be subdivided into three main categories based on their molecular weight and solubility: (i) THMWDF that can be subdivided into soluble and insoluble high-molecular-weight DF (IHMWDF and SHMWDF); (ii) resistant starch (RS), which is classified into five different subcategories (RS1-RS5); and (iii) low-molecular-weight DF (LMWDF). The main constituents of THMWDF are represented by cellulose, lignin, hemicellulose, insoluble pectin (IHMWDF), and hydrocolloids such as guar gum, soluble pentosan, and soluble pectin (SHMWDF). The name RS indicates any starch that is resistant to hydrolysis by the enzymes of the small intestine. Finally, the third group, represented by the water-soluble and 78% ethanol-soluble DF fraction called LMWDF, comprises the various prebiotics such as inulin, fructose oligosaccharide, and galactose oligosaccharide. The AOAC method 991.43 [43] has been widely used over the past four decades as the gold standard for DF measurement. However, it is not suitable for the determination of all components of DF as defined by CODEX Alimentarius, as it only permits the measurement of THMWDF and certain types of RS, but not the content of LMWDF constituents [63]. For this reason, in addition to the measurement of THMWDF using the method of Lee and Prosky [43], we determined the amount of inulin in the heads, which is known to represent the prevalent LMWDF component in artichoke [64,65]. In recent decades, a number of studies have demonstrated the value of inulin as a promising functional nutrient due to its wide range of biological activities, acting as a prebiotic to improve the intestinal microbe environment, or as an antioxidant or anticancer, or regulating blood sugar and blood fat, or promoting mineral and vitamin absorption [66].

THMWDF content of heads differed significantly among the analysed genotypes, ranging from 5.75 to 7.10 g 100 g^−1^ FW, with Orte 2 genotypes showing significantly higher values than Orte 1 genotypes and the landraces/clones used as controls (Table 5). In all genotypes, the IHMWDF was the prevalent fraction of the THMWDF, ranging from 57% (“Grato 1”) to 71% (“Castellammare” and “C3”), with the Orte 2 genotypes exhibiting the highest absolute values (Table 5). Furthermore, the head inulin content of the genotypes varied significantly, ranging from 2.25 to 5.12 g 100 g^−1^ FW (corresponding to 11.6 and 26.25 g 100 g^−1^ DW). Once again, Orte 2 genotypes displayed significantly higher values than the Orte 1 genotypes and the landraces/clones used as controls (Table 5). These results indicated that particular attention should be devoted to the genotypes belonging to the Orte 2 population that could have an added nutritional value if consumed fresh by considering their high THMWDF and inulin contents. Di Venere et al. [67], analysing 35 cultivars/clones of different origins and varietal typology, showed remarkable differences in inulin content of heads, in a range between 1 and 6 g 100 g^−1^ FW, with approximately 70% of the genotypes showing values between 3 and 5 g 100 g^−1^ FW. In contrast, Lattanzio et al. [6] and Pandino et al. [4] reported higher values for head inulin content expressed on a dry-matter basis (18–36 g 100 g^−1^ DW) compared to those obtained in this study. These differences could be due to the samples used for analysis in the different studies, which may not be at the same physiological stage during the harvest phase, as well as to the different genotypes analysed and the environmental conditions in which they were grown [6,68].

The range of mineral content of the heads in the studied genotypes (Appendix A) is within the range reported in the literature [7,18,69,70]. In all the studied genotypes, K was the most abundant mineral, as has been previously reported [7,18,69,70]. Indeed, it has been documented previously that K is the nutrient most readily taken by artichoke plants during the growing cycle [71]. With the exception of Fe and Zn, the content of the other minerals was significantly lower in Orte 2 genotypes than in Orte 1 genotypes and the landraces/clones used as controls (Appendix A), in agreement with the lowest ash content found in the two genotypes belonging to the Orte 2 population (Table 5). Regarding the Ca content of heads, it is important to emphasize the significantly higher values found in Orte 1 genotypes and in the accession of the landrace “Campagnano”. The high Ca content of these genotypes can be considered an important quality trait, as it is related to improved storage qualities of fruits and vegetables and could be exploited to extend the storage life of artichoke heads and enhance the quality of the final product during and following storage [7].

The total phenolic content (TPC) and total flavonoid content (TFC) of the heads of the studied genotypes are reported in Appendix A. TPC ranged from 70.46 to 131.1 mg GAE g^−1^ DW, whereas TFC ranged from 3.3 to 5.37 mg RUE g^−1^ DW, matching with the data reported in the literature [4,7,21]. Orte 2 genotypes showed the highest polyphenol and flavonoid contents, followed by Orte 1 genotypes, “C3”, and “Campagnano”, whereas “Grato 1” and “Castellammare” exhibited the lowest TPC and TFC values (Appendix A). It is widely recognized that polyphenols play a key role in human nutrition, since their regular consumption has been associated with a reduced risk of several chronic diseases, including cancer, cardiovascular disease, and neurodegenerative disorders [72]; therefore, phenolic-rich foods are highly appreciated by consumers. Globe artichoke has been found to be a substantial dietary source of 5-O-caffeoylquinic acid (chlorogenic acid), 1,5 di-O-caffeoylquinic acid (cynarin), and the flavonoid apigenin-7-O-glucuronide, which are considered the most important compounds involved in the many therapeutic properties attributable to artichoke, such as hepaprotective, antioxidative, anticarcinogenic, urinative, antibacterial, antiglycaemic, and anticholesterol activities [6]. In the context of processed foods, polyphenols are significant due to their involvement in many oxidative reactions that take place during the processing stage. In particular, polyphenols are responsible for the degradation of food flavour, colour, and a number of other sensory characteristics [4]. Among the studied genotypes, those belonging to the Orte 2 population appear to be well suited for fresh consumption due to their high total phenolic and flavonoid content (Appendix A), a feature that could be highly valued by the market. In contrast, “Grato 1” and “Castellammare”, which have low polyphenol and flavonoid contents, could be more suitable for the processed food industry. Indeed, as discussed by Lattanzio et al. [6], browning phenomena, due to polyphenol oxidation, could be reduced by using genotypes with a low polyphenol content. The analysis of the phenolic profile of the flower heads of the two groups of “Carciofo Ortano” genotypes will be the focus of a future study to determine their optimal end use (fresh consumption, processing, or pharmaceutical industry).

## 4. Conclusions

This study focused on the molecular, morphological, and nutritional characterisation of the artichoke “Carciofo Ortano,” a landrace at risk of genetic erosion still grown in the plains along the Tiber River in the area of the Orte municipality in the Lazio region (Central Italy). Molecular analysis based on SSR and ISSR markers allowed the univocal identification of the representative genotypes of the “Carciofo Ortano” landrace, confirming their belonging to the “Romanesco” varietal typology, and revealed a high level of genetic variability within this landrace. Indeed, clustering analyses based on molecular marker data indicated the presence of two distinct populations within the landrace. The first, named Orte 1, is genetically closer to the “Campagnano” and “Castellammare” landraces, and the clones derived from them, and mainly cultivated in the coastal areas of the Lazio region; the other, called Orte 2, is closer to the “Montelupone” landrace, and cultivated in the district of the homonymous municipality in the province of Macerata (Marche, Italy). These results suggest that in the investigated area, several introductions of different genotypes, likely from distinct geographical areas, could have taken place, although this is not apparent from historical surveys and interviews with farmers. Despite the high level of within-population genetic variation detected, the two populations were clearly differentiated each to other. This could be the consequence of a limited exchange of material between the farmers, as evidenced by the fact that the two populations are rarely grown in mixed cultivation on the same farm. The detected genetic variability, on the one hand, provides useful information for the implementation of appropriate in situ conservation strategies and, on the other hand, emphasizes the need for clonal selection programs to provide farmers with more uniform and selected material, especially in view of a revival of the cultivation of this landrace. According to molecular analyses, the genotypes belonging to the two populations were also differentiated for some morphological and nutritional traits. Based on the morphological evaluations, the Orte 1 genotypes showed agronomic profiles with useful values of earliness and production (mainly weight of heads) compared to those of Orte 2 genotypes. However, the presence in the landrace of genotypes with a different maturity stage of the flower heads could allow farmers to respond for a longer period to local market demand (more than one month). The nutritional and chemical characterisation also provided useful information about the potential specific end-use of the final product of the genotypes of the two populations. Orte 1 genotypes, with a lower content of both polyphenols and inulin and the higher Ca content, could be more suitable for industrial processing, owing to their potential lower aptitude to browning phenomena and white precipitate during processing operations and cold storage. On the other hand, because of their higher content of polyphenols, inulin, and THMWDF, Orte 2 genotypes appear to be better suited for fresh consumption.

The morphological and molecular data obtained in this study for the “Carciofo Ortano” can assist the landrace protection schemes that are currently being developed in Italy. Indeed, these data were useful for the inclusion of the “Carciofo Ortano” landrace in the Lazio Regional Register established by the Regional Law no. 15/2000 (Protection of autochthonous genetic resources of agricultural interest) and could facilitate the registration of the landrace in the National Register of Biodiversity managed by the Italian Ministry of Agriculture, Food and Forestry (Ministerial Decree no. 38,654 of 4 November 2019 and Ministerial Decree no. 13,073 of 17 April 2020). Moreover, the availability of nutritional values and chemical composition of heads is essential for understanding the potential benefits of “Carciofo Ortano” on human health, as well as supporting its possible production and marketing. All these activities are important not only for the future survival of the landrace in situ, but also could be the basis for the revival of its cultivation on a medium/large scale, given the renewed interest in this vegetable by local farmers.

## Figures and Tables

**Figure 1 plants-12-01844-f001:**
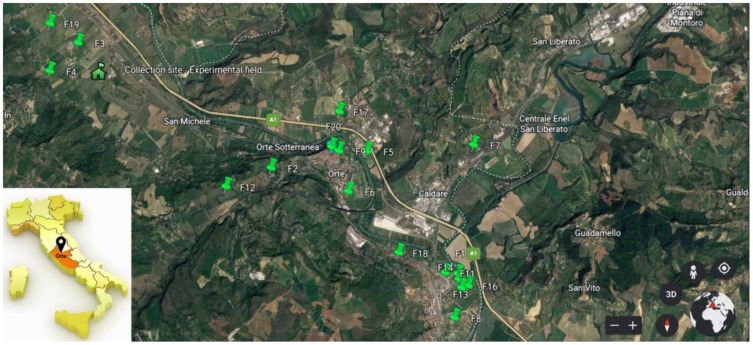
Map of the smallholdings and family gardens (F1–F20) in the municipality of Orte where the artichoke plantings from which the plant material was collected for the molecular analysis are located. The location of the collection experimental field is also reported (adapted from Google Maps).

**Figure 2 plants-12-01844-f002:**
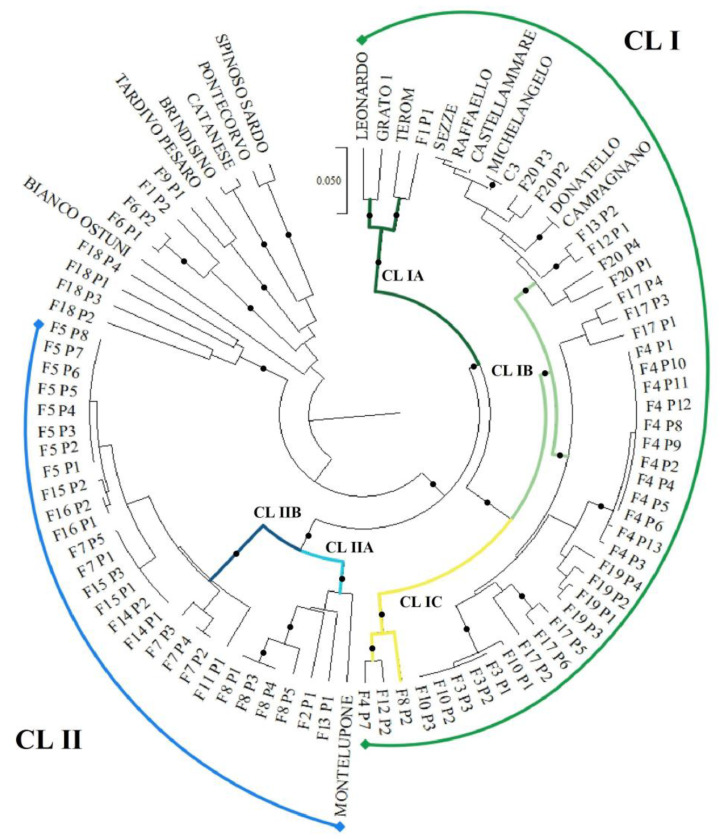
UPGMA dendrogram of the genetic relationships among the 90 artichoke genotypes (73 plants sampled in the Orte municipality and 17 accessions of landraces/clones belonging to the 4 varietal types cultivated in Italy) generated by Nei’s coefficient [50] using both SSR and ISSR molecular markers. Branches indicated with dots represent bootstrap support more than 80% (1000 repetitions).

**Figure 3 plants-12-01844-f003:**
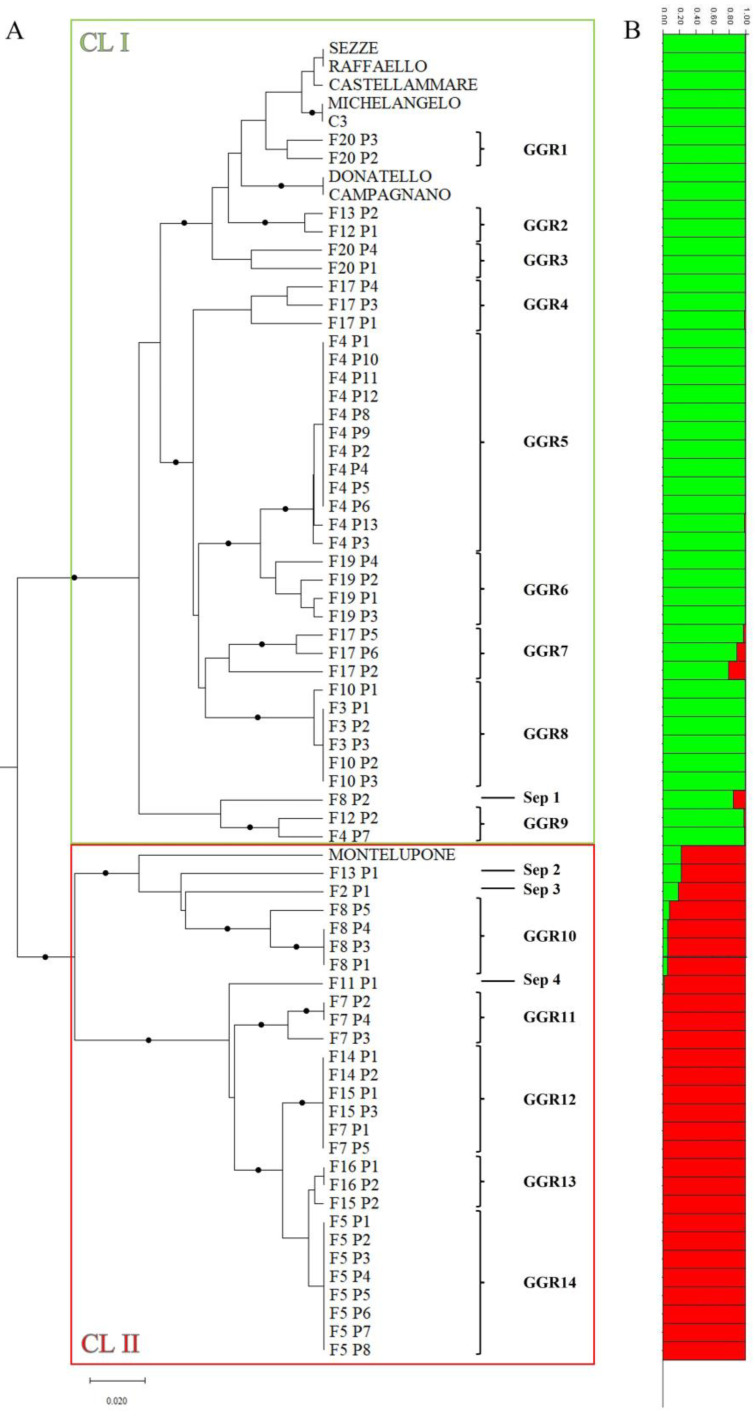
Clustering of 72 artichoke genotypes (64 plants sampled in the Orte municipality and 8 accessions belonging to the “Romanesco” type). (**A**) UPGMA dendrogram based on Nei’s coefficient [50] among the 72 genotypes. Branches indicated with dots represent bootstrap support of more than 80% (1000 repetitions). (**B**) Structure bar plots of average proportion membership (Q) for the same genotypes for K = 2 (in green and in red). The UPGMA dendrogram indicates the distinct genetic groups (GGR 1–14) and the four plants genetically separated from them (Sep 1–4) used to identify the plants for the establishment of the collection experimental field.

**Figure 4 plants-12-01844-f004:**
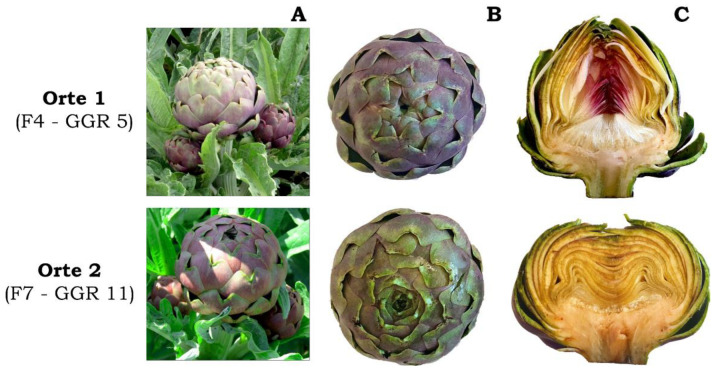
Flower heads from representative genotypes of the two populations (Orte 1 and Orte 2) of the landrace “Carciofo Ortano”. Primary and secondary flower heads in the field plants (**A**), primary flower heads seen from above (**B**), and their longitudinal section (**C**).

**Figure 5 plants-12-01844-f005:**
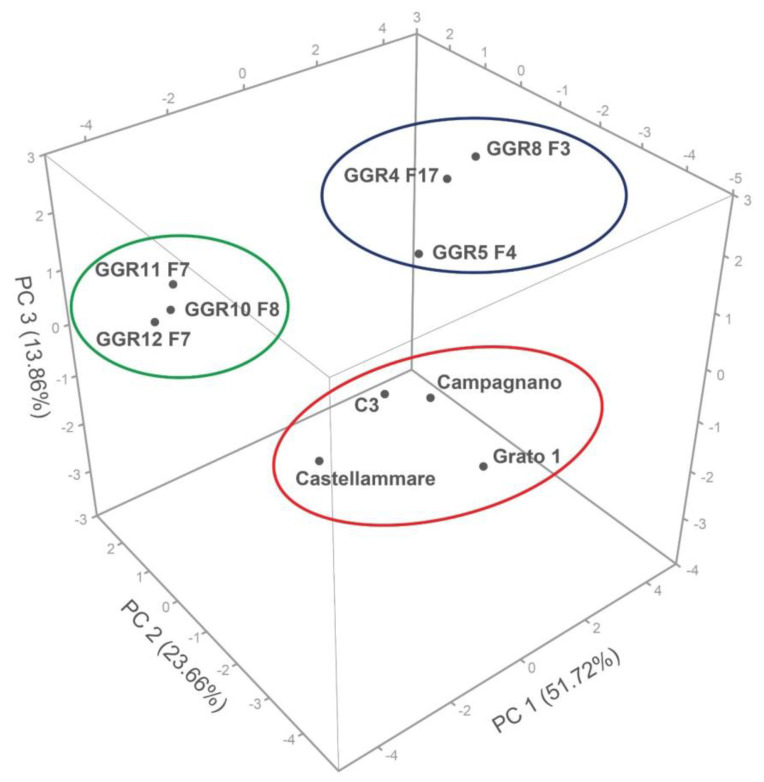
Distribution of the analysed genotypes against the first three principal component functions. The genotypes of Orte 1 and Orte 2 populations are circled in blue and in green, respectively, whereas the four landraces/clones used as reference genotypes are in red.

**Table 1 plants-12-01844-t001:** Genetic diversity parameters from the 9 SSR and the 6 ISSR markers used for the analysis of 90 artichoke genotypes (73 plants sampled in the Orte municipality and 17 accessions of landraces/clones belonging to the 4 varietal types cultivated in Italy). Major Allele Frequency (MAF); Expected Heterozygosity (He); Observed Heterozygosity (Ho); Polymorphism Information Content (PIC); Total Bands (TB); Polymorphic Bands (PB); Polymorphism percentage (Pol %).

SSR Marker	Allele No			MAF	He	PIC	Ho
CDAT_02	7			0.505	0.635	0.577	0.374
CELMS_08	7			0.703	0.485	0.465	0.978
CDAT_04	8			0.495	0.658	0.609	0.429
Cmal_06	4			0.879	0.221	0.212	0.934
CELMS_05	9			0.505	0.637	0.581	0.912
Cmal_08	4			0.912	0.165	0.161	0.967
CDAT_01	8			0.505	0.646	0.594	0.989
CELMS_40	10			0.527	0.650	0.609	0.912
CsPal_03	8			0.527	0.620	0.563	0.89
TOT	65						
MEAN	7.22			0.618	0.524	0.486	0.821
ST. DEV.				0.17	0.195	0.176	0.24
ISSR Marker	TB	PB	% Pol	MAF	He	PIC	
ISSR_24	17	17	100	0.789	0.29	0.236	
ISSR_28	11	7	64	0.979	0.04	0.039	
UBC_840	15	15	100	0.870	0.200	0.170	
UBC_848	10	10	100	0.699	0.363	0.282	
UBC_855	12	12	100	0.803	0.265	0.213	
UBC_857	11	11	100	0.783	0.292	0.235	
TOT	76	72					
MEAN	12.67	12	94	0.821	0.242	0.196	
ST. DEV.				0.087	0.102	0.078	

**Table 2 plants-12-01844-t002:** Genetic diversity parameters of SSR and ISSR markers for the “Carciofo Ortano” genotypes classified into two groups according to model-based cluster analysis. N: number of individuals for each group; Ne: number of effective alleles; Npa: number of private alleles; I: Shannon index; He: expected heterozygosity; Ho: observed heterozygosity. The asterisk indicates significant differences at *p* ≤ 0.01.

SSR Marker						
	N	Ne	Npa	I	He	Ho
Orte 1	37.00	1.78	4.00	0.54	0.39	0.78
Orte 2	27.00	2.20	11.00	0.82	0.54	1.00
*p*-value Kruskal–Wallis Test		<0.01 *		<0.01 *	<0.01 *	>0.01
ISSR Marker						
	N	Ne	Npa	I	He	
Orte 1	37.00	1.39	15.00	0.35	0.23	
Orte 2	27.00	1.36	6.00	0.28	0.20	
*p*-value Kruskal–Wallis Test		>0.01		>0.01	>0.01	

**Table 3 plants-12-01844-t003:** Analysis of molecular variance for the partitioning of SSR and ISSR marker diversity of “Carciofo Ortano” genotypes classified into two groups according to model-based cluster analysis. *p*(Φ)–Φ-statistical probability level after 9999 permutations.

SSR Marker							
	df	SS	MS	Est. Var.	%	Φ-statistic	*p*(Φ)
Among Pops	1.00	71.408	71.408	2.596	80.55%	0.905	<0.001
Within Pops	62.00	38.889	0.627	0.627	19.45%		
Total	63.00	110.297		3.223	100%		
ISSR Marker							
	df	SS	MS	Est. Var.	%		
Among Pops	1.00	106.87	106.87	3.23	34.94%	0.349	<0.001
Within Pops	62.00	372.93	6.01	6.01	65.06%		
Total	63.00	479.80		9.25	100.00%		

**Table 4 plants-12-01844-t004:** Differences in twelve morphological quantitative traits (means of six measurements) of the genotypes of Orte 1 (GGR4, GGR5, and GGR8) and Orte 2 (GGR10, GGR11, and GGR12) populations and the four landraces/clones belonging to the “Romanesco” type. PH: Plant height (cm); MSD: Main stem diameter (cm); DHL: Distance between the youngest leaf on the main floral stem and the central flower-head (cm); LCH: Length of central head (cm); L/D ratio: Length/Diameter ratio of central head (cm/cm); RECD: Diameter of receptacle (cm); WCH: Weight of the central head (g); LH1: Length of the first lateral head on the lateral shoots (cm); L/D ratio1: Length/Diameter ratio of first lateral head (cm/cm); WH1: Weight of the primary head (g); Y: Total production (g); THA: Time of central head appearance (days). Different letters indicate statistically significant differences among genotypes at *p* ≤ 0.05 (ANOVA analysis, Tukey test); *** indicate significant differences at *p* ≤ 0.001. CV: coefficient of variation expressed as percentage.

	PH	MSD	DHL	LCH	L/D Ratio	RECD	WCH	LH1	L/D Ratio1	WH1	Y	THA
Campagnano	66.833 ^b^	2.316 ^bc^	24.750 ^ab^	90.000 ^a^	0.947 ^b^	4.566 ^ab^	260.558 ^abc^	5.333 ^a^	1.011 ^b^	81.240 ^bc^	692.798 ^bc^	102.333 ^b^
Castellammare	52.833 ^d^	2.621 ^a^	20.450 ^c^	88.500 ^a^	0.938 ^b^	4.666 ^ab^	245.468 ^abc^	5.066 ^ab^	0.996 ^b^	71.316 ^c^	617.913 ^cd^	95.000 ^c^
C3	58.666 ^cd^	2.666 ^a^	20.666 ^c^	96.833 ^a^	0.973 ^b^	4.983 ^a^	287.665 ^ab^	5.350 ^a^	0.983 ^b^	94.765 ^a^	798.718 ^a^	84.166 ^d^
Grato 1	61.666 ^bc^	2.200 ^c^	22.916 ^bc^	96.000 ^a^	1.115 ^a^	4.000 ^b^	230.485 ^bc^	5.350 ^a^	1.173 ^a^	76.138 ^c^	597.956 ^d^	103.833 ^b^
Orte1-GGR4 F17	73.000 ^a^	2.533 ^ab^	27.000 ^a^	98.166 ^a^	0.960 ^b^	5.150 ^a^	303.815 ^a^	5.783 ^a^	0.991 ^b^	91.836 ^ab^	768.151 ^a^	105.000 ^b^
Orte1-GGR5 F4	67.000 ^b^	2.483 ^ab^	23.166 ^bc^	95.666 ^a^	0.953 ^b^	4.983 ^a^	304.283 ^a^	5.716 ^a^	1.028 ^b^	93.245 ^a^	785.028 ^a^	105.666 ^b^
Orte1-GGR8 F3	74.333 ^a^	2.316 ^bc^	24.333 ^ab^	95.000 ^a^	0.955 ^b^	5.16 ^a^	285.355 ^ab^	5.366 ^a^	1.016 ^b^	90.566 ^ab^	724.255 ^ab^	103.333 ^b^
Orte2-GGR12 F7	53.333 ^d^	2.383 ^bc^	20.000 ^c^	72.666 ^b^	0.805 ^c^	4.800 ^a^	231.573 ^bc^	4.300 ^c^	0.876 ^c^	72.033 ^c^	589.273 ^d^	118.000 ^a^
Orte2-GGR11 F7	54.500 ^d^	2.403 ^bc^	20.333 ^c^	74.833 ^b^	0.790 ^c^	4.850 ^a^	238.103 ^bc^	4.533 ^bc^	0.865 ^c^	74.323 ^c^	608.093 ^d^	118.000 ^a^
Orte2-GGR10 F8	54.000 ^d^	2.400 ^bc^	21.000 ^c^	74.000 ^b^	0.801 ^c^	4.716 ^a^	222.413 ^c^	4.416 ^bc^	0.885 ^c^	73.206 ^c^	588.620 ^d^	118.333 ^a^
CV	13.368	5.947	10.497	11.724	10.823	7.099	12.159	10.336	9.305	11.829	12.769	10.301
*p*	***	***	***	***	***	***	***	***	***	***	***	***

**Table 5 plants-12-01844-t005:** Nutritional values expressed as g 100 g^−1^ FW (means of six analyses) of the heads of the eight studied artichoke genotypes. HMWDF: high-molecular-weight dietary fibre; IDF: insoluble dietary fibre; SDF: soluble dietary fibre. Different letters indicate statistically significant differences among genotypes at *p* ≤ 0.05 (ANOVA analysis, Tukey test); *** indicate significant differences at *p* ≤ 0.001; * indicates significant differences at *p* ≤ 0.05; ns indicate no significant differences. CV: coefficient of variation expressed as percentage.

	Moisture	Ash	Proteins	Lipids	Carbohydrates	THMWDF	IHMWDF	SHMWDF	Inulin
Campagnano	82.10 ^b^	1.51 ^a^	2.96 ^b^	0.23 ^a^	7.13 ^b^	6.08 ^b^	4.03 ^c^	2.05 ^bc^	3.30 ^b^
Castellammare	81.53 ^c^	1.45 ^ab^	3.25 ^a^	0.22 ^a^	7.30 ^b^	6.25 ^b^	4.43 ^b^	1.82 ^cd^	2.93 ^c^
C3	83.38 ^a^	1.45 ^ab^	3.02 ^ab^	0.23 ^a^	6.17 ^c^	5.75 ^c^	4.1 ^c^	1.67 ^d^	2.92 ^c^
Grato 1	80.88 ^d^	1.40 ^b^	3.18 ^ab^	0.25 ^a^	7.35 ^b^	6.93 ^a^	3.63 ^c^	2.70 ^a^	2.25 ^d^
Orte1-GGR4 F17	82.96 ^a^	1.36 ^b^	3.06 ^ab^	0.22 ^a^	6.32 ^c^	6.07 ^b^	3.87 ^c^	2.20 ^b^	3.33 ^b^
Orte1-GGR5 F4	83.33 ^a^	1.35 ^b^	3.00 ^ab^	0.23 ^a^	5.97 ^c^	6.12 ^b^	3.85 ^c^	2.27 ^b^	3.57 ^b^
Orte2-GGR10 F8	80.50 ^de^	1.20 ^c^	3.03 ^ab^	0.22 ^a^	7.98 ^a^	7.07 ^a^	4.94 ^a^	2.13 ^b^	5.12 ^a^
Orte2-GGR11 F7	80.25 ^e^	1.21 ^c^	3.00 ^ab^	0.23 ^a^	8.20 ^a^	7.10 ^a^	4.93 ^a^	2.17 ^b^	5.05 ^a^
CV	1.55	8.21	3.27	4.33	11.77	8.22	11.75	14.49	28.69
*p*	***	***	*	ns	***	***	***	***	***

## Data Availability

The data contained within the present article and in its Appendix A are freely available upon request to the corresponding author.

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
