# Peer review of "Morphological, Molecular, and Nutritional Characterisation of the Globe Artichoke Landrace “Carciofo Ortano”"

_plants, 2023, doi:10.3390/plants12091844_

Round 1

Reviewer 1 Report

It is pretty hard to understand the content of this paper. The abstract, introduction, results and conclusion is too long and too complex (not well organized). Many of the contents can be removed. The readers can not catch the major points of this research. 

(1) It is far from enough to use only 12 SSR and 10 ISSR to inverstigate the molecular diversity. It is easy to design plentiful and diverse molecular markers by a simple transcriptome and/or sequnencing. 

(2)There is no information on the chromosome locations of the SSR and ISSR markers used. Please privide the full name of SSR and ISSR when first use.

(3) Is the globe artichoke self or cross - pollination? 

(4) "in situ" should be italic.

(5) Line 215: what is"FAM, TAMRA and JOE"?

(6) It is better to provide pictures for these plant marterials used.

Author Response

Response to Reviewer 1 Comments

It is pretty hard to understand the content of this paper. The abstract, introduction, results and conclusion is too long and too complex (not well organized). Many of the contents can be removed. The readers can not catch the major points of this research. 

General Response: We thank the Reviewer 1 for the general comment and criticism on the manuscript. Following his/her suggestions, we have significantly revised the text of the manuscript with regard to “abstract”, “introduction”, “materials and methods”, and “results and discussion” sections, in particular eliminating some repetitions and information that is not relevant to the overall objectives of the work. However, we have not modified the “conclusions” section, as in our opinion it summarises the most important results obtained in the work and its importance for the conservation of the landrace and for its future revival. We hope that on the basis of the changes made, the fluidity and consequentiality of the information will be clearer so that the reader, as requested by the reviewer, can fully focus on the results obtained in this work.

Point 1: It is far from enough to use only 12 SSR and 10 ISSR to inverstigate the molecular diversity. It is easy to design plentiful and diverse molecular markers by a simple transcriptome and/or sequnencing.

Response 1: We are aware that it is possible to develop numerous molecular markers through new sequencing technologies, but often the choice of the number and type of molecular markers depends on the objectives of the study and also on the available economic budget. One of the objectives of our study was to develop a molecular-genetic methodology for the study of the genetic structure of the landrace 'Carciofo Ortano' in order to accurately define the most suitable strategies to be adopted for its conservation in situ, and for its unambiguous identification with respect to landraces/clones or varieties belonging to the four varietal types cultivated in Italy, namely Romaneschi, Spinosi, Violetti and Catanesi. To achieve this goal, based on the available budget, 12 SSR and 10 ISSR markers were selected according to their ability to detect a high level of polymorphism (PIC value) and, in particular, to distinguish among the four varietal types cultivated in Italy, as evidenced by numerous previous genetic studies of globe artichoke germplasm collections. On the basis of the results obtained the already developed and therefore low-cost SSR and ISSR markers were highly informative for the achievement of the intended goal because, as illustrated in the manuscript, allowed the univocal identification of the representative genotypes of the “Carciofo Ortano” landrace, confirming their belonging to the “Romanesco” varietal typology, and revealed a high level of genetic variability within this landrace. Indeed, clustering analyses based on molecular marker data indicated the presence of two distinct populations within the landrace. Despite the high level of within population genetic variation detected, the two populations were genetically differentiated each to other and from the landraces/clones of the main varietal types cultivated in Italy. The detected genetic variability, on the one hand, provided useful information for the implementation of appropriate in situ conservation strategies and, on the other hand, emphasized the need for clonal selection programs to provide farmers with more uniform and selected material, especially in view of a revival of the cultivation of this landrace. In conclusion, although we are aware that nowadays the exploration of plant biodiversity is boosted by advanced sequencing technologies, which provide the opportunity to simultaneously discover and test a high number of molecular markers for the analysis of plant genetic diversity on a genome-wide scale, it is our opinion that on the basis of the results obtained and their comparison with data reported in literature, the low-cost 12 SSR and 10 ISSR markers used in this study were sufficiently informative for the analysis of the genetic variability and structure of the local variety 'Carciofo Ortano'.

Point 2: There is no information on the chromosome locations of the SSR and ISSR markers used. Please provide the full name of SSR and ISSR when first use.

Response 2: As indicated in the manuscript, in particular in Table S3, the 12 selected SSR markers were developed by Acquadro et al [reference no. 35-38] and Sonnante et al [39], where their assigned names, GeneBank accession number, and all characteristics, including, where determined, their chromosome location, are given. As reported in the text of the paper, these SSR markers have been used in previous genetic studies of several globe artichoke germplasm collections using the original names assigned to them, as in our study. To satisfy the reviwer’s request, in Table S3 is reported the chromosome location of nine of the twelve SSRs used in our study (LG).  

Due to the nature of ISSR markers, it is not possible to indicate their chromosomal location. Indeed, ISSRs are DNA fragments of about 100-3000 bp located between adjacent, oppositely oriented microsatellite regions. ISSRs are amplified by PCR using microsatellite core sequences as primers with a few selective nucleotides as anchors into the non-repeat adjacent regions (16-18 bp). About 10-30 fragments from multiple loci are generated simultaneously, separated by gel electrophoresis and scored as the presence or absence of fragments of particular size.

Point 3: Is the globe artichoke self or cross - pollination?

Response 3: As stated in the text of the paper (lines 443-445 of the old version of the manuscript) the globe artichoke is a higly outcrossed species, which is mainly propagated vegetatively by using the “carducci” (basal shoots) or the “ovoli” (semi-dormant shoots with a limited root system). Albeit selfing is not precluded in globe artichoke, cross-ferilization is promoted by protandry (Mauromicale and Ierna, Agronomie 20, 197-204, 2000). However, to take into account the reviewer's comment we have reported in the first sentence of the “introduction” section that the globe artichoke is an open pollinated specie (see lines 34-36 of the present revised version).

Point 4: "in situ" should be italic.

Response 4: The suggested correction has been made throughout the text.

Point 5: Line 215: what is "FAM, TAMRA and JOE"?

Response 5: FAM, TAMRA and JOE are fluorescent dyes used to detect PCR-amplified fragments separated by capillary electrophoresis. This has now been specified in the text (see line 184 of the present revised version of the manuscript).

Point 6: It is better to provide pictures for these plant marterials used.

Response 6: Following the suggestion from the Reviewer 1, we have included two figures showing pictures of the plant material used in this study. The first (Figure S2) concerns the pictures of some plants selected in the 20 farms for the genetic/molecular analysis. The second (Figure S5) includes the pictures of the main heads of some plants of the two populations identified in the “Carciofo Ortano” landrace growth in the the collection experimental field and used in the morphological analyses.

Reviewer 2 Report

This manuscript entitled “Recovery, morphological, molecular, and nutritional characterization of the globe artichoke “Carciofo Ortano”, a landrace with a high risk of genetic erosion cultivated in Central Italy” describes the recovery and characterization of a landrace of globe artichoke which is at high risk of genetic erosion and is traditionally cultivated in central Italy. The study involved morphological, molecular, and nutritional analysis to determine the plant's characteristics and potential use. The paper is well written. However, the authors should further improve the quality in the sense that all statements are clear enough to the readers and redundancy is minimized throughout the text. In addition to these general comments, below are the specific comments about the changes necessary to the text.

Abstract

This section is relatively long. Please briefly summarize your study mentioning the most important findings and perspectives.

Introduction

For artichokes production numbers, it is important to provide very recent updated information. It would be great if the authors provide 2022 production values instead of 2020.

Material and Methods

The experiments are well implemented and executed. There are too many subtitles, I would suggest binding the close ones together.

I see Figure S1 is important to be highlighted as the main figure rather than supplementary.

A table of primers used in the molecular analysis would be valuable in this section.

For the morphological analyses, did you take some photos that could be relevant to this research? If so, please include them as main or supplementary material.

In the Data Analysis sub-section, replace the sub-title “Molecular analysis” with “Molecular data”. The same goes for “Morphological analysis” with “Morphological data”, and “Nutritional and chemical data”

Results and discussion

Figures presented in this paper should be inserted in high quality.

Conclusion

I would recommend including future study perspectives for possible research outcomes.

Author Response

Response to Reviewer 2 Comments

This manuscript entitled “Recovery, morphological, molecular, and nutritional characterization of the globe artichoke “Carciofo Ortano”, a landrace with a high risk of genetic erosion cultivated in Central Italy” describes the recovery and characterization of a landrace of globe artichoke which is at high risk of genetic erosion and is traditionally cultivated in central Italy. The study involved morphological, molecular, and nutritional analysis to determine the plant's characteristics and potential use. The paper is well written. However, the authors should further improve the quality in the sense that all statements are clear enough to the readers and redundancy is minimized throughout the text. In addition to these general comments, below are the specific comments about the changes necessary to the text.

General Response: Following the suggestion from the Review 2, we revised the text of the manuscript with regard “introduction”, “materials and methods”, and “results and discussion” sections, in particular eliminating some repetitions and information that is not relevant to the overall objectives of the work. We hope that on the basis of the changes made, the fluidity and consequentiality of the information will be clearer and the general quality of the manuscript will be improved.

Point 1: Abstract. This section is relatively long. Please briefly summarize your study mentioning the most important findings and perspectives. 

Response 1: Following the suggestion from the Review 2, the “abstract” section has been significantly shortened by focusing on the main results obtained and their importance for the conservation of the landrace.

Point 2: Introduction. For artichokes production numbers, it is important to provide very recent updated information. It would be great if the authors provide 2022 production values instead of 2020.

Response 2: We thank the Reviewer for the comment. The two databases consulted regarding world (FAOSTAT) and national (Italy) (ISTAT) artichoke production have recently been updated (March 2023), but the most recent data relate to production and cultivated area in 2021. Consequently, the data in the text of the manuscript have been updated to 2021.

Point 3: Material and Methods. The experiments are well implemented and executed. There are too many subtitles, I would suggest binding the close ones together.

Response 3: Following the suggestion from the Review 2, we have reduced the number of subheadings where possible in the “materials and methods” section (see e.g. point 2.2. “Molecular characterization of Carciofo Ortano landrace” and the point 2.5. “Nutritional and chemical analysis of primary flower heads”).    

Point 4: Material and Methods. I see Figure S1 is important to be highlighted as the main figure rather than supplementary.

Response 4: We thank the reviewer for the suggestion. However, considering the length of the article and the already large number of Figures and Tables included in the original text, it is our opinion, in order not to burden the manuscript further, that the figure in question remain as an additional figure, even if we are of course ready to follow the Reviewer’s recommendation in case he/she maintain his/her request. 

Point 5: Material and Methods. A table of primers used in the molecular analysis would be valuable in this section.

Response 5: We thank the reviewer for the suggestion. However, as stated before, considering the length of the article and the already large number of Figures and Tables included in the original text, it is our opinion, in order not to burden the manuscript further, that it is not necessary to insert a main Table for the SSR and ISSR primers in the “materials and methods” section. On the other hand, the characteristics of the SSR and ISSR primers used in the molecular analysis are shown in Tables S3 and S4.   

Point 6: Material and Methods. For the morphological analyses, did you take some photos that could be relevant to this research? If so, please include them as main or supplementary material.

Response 6: Following the suggestion from the Reviewer 2, we have included two figures showing pictures of the plant material used in this study. The first (Figure S2) concerns the pictures of some plants selected in the 20 farms for the genetic/molecular analysis. The second (Figure S5) includes the pictures of the main heads of some plants of the two populations identified in the “Carciofo Ortano” landrace growth in the the collection experimental field and used in the morphological analyses.

Point 7: Material and Methods. In the Data Analysis sub-section, replace the sub-title “Molecular analysis” with “Molecular data”. The same goes for “Morphological analysis” with “Morphological data”, and “Nutritional and chemical data”.

Response 7: Following the suggestion of the Reviewer 2, in the Data Analysis sub-section the title of subheadings “Molecular analysis”, “Morphological analysis”, and “Nutritional and chemical analysis of primary flower heads” have been replaced with “Molecular data”, “Morphological data”, and “Nutritional and chemical data”, respectively.

Point 8: Results and Discussion. Figures presented in this paper should be inserted in high quality.

Response 8: Following the suggestion of the Reviewer 2, the old Figures (330 dpi) were replaced with high quality Figures (1200 dpi).

Point 9: Conclusion. I would recommend including future study perspectives for possible research outcomes.

Response 9: We thank the reviewer for the above comment. However, it is our opinion that the conclusions reported, after summarising the main results obtained and their importance for the conservation and possible revival of the landrace, also highlight possible implications for future studies. For example, as reported in the text of the manuscript, the detected genetic variability through molecular analysis, on the one hand, provides useful information for the implementation of appropriate in situ conservation strategies and, on the other hand, emphasizes the need for clonal selection programs to provide farmers with more uniform and selected material, especially in view of a revival of the cultivation of this landrace. Furthermore, we also highlighted that the analysis of the phenolic profile of the flower heads of the two groups of “Carciofo Ortano” genotypes will be the focus of a future study to determine their optimal end use (fresh consumption, processing, or pharmaceutical industry).

Reviewer 3 Report

The title of the article is too long

Suggested:

Morphological, molecular, and nutritional characterization of the globe artichoke landrace “Carciofo Ortano”

The abstract exceeds the number of words allowed

Redaction needs some improvement

What is FAO, FAOSTAT Crops, 2020?

SISTAT? Or ISTAT 2009?

Complete name of metric units should be written when first mentioned (Kt, Mt)

Table S1               the table should be self-explanatory. What does the PI mean?

Number all tables and figures in order as they are mentioned

some materials and methods are in the results section and viceversa, example:

3.4. Nutritional and chemical characterization

These are materials and methods

Three plants for each of the three replicates for four different genetic groups, two 743 belonging to the Orte 1 population (GGR4 and GGR5) and the others to the Orte 2 popu-744 lation (GGR10 and GGR11), and nine plants for each of the four landraces/clones used as 745 reference genotypes (three plants per replicate) were selected for the determination of nu-746 tritional value and chemical composition of their primary flower heads.

Author Response

Response to Reviewer 3 Comments

Point 1: The title of the article is too long. Suggested: Morphological, molecular, and nutritional characterization of the globe artichoke landrace “Carciofo Ortano”.

Response 1: We thank the reviewer for the suggestion. The suggested correction has been made in the title.

Point 2: The abstract exceeds the number of words allowed.

Response 2: Following the suggestion from the Reviewer 3, the “abstract” section has been significantly shortened by focusing on the main results obtained and their importance for the conservation of the landrace.

Point 3: Redaction needs some improvement

Response 3: Following the suggestion from all the Reviewers, we revised the text of the manuscript with regard “introduction”, “materials and methods”, and “results and discussion” sections, in particular eliminating some repetitions and information that is not relevant to the overall objectives of the work. We hope that on the basis of the changes made, the fluidity and consequentiality of the information will be clearer and the general quality of the manuscript will be improved. We are aware that if the proposed corrections will be accepted by the reviewers, the formatting of the text of the manuscript and the correct inclusion of the Figures and Tables in the final text will have to be revised and improved. However, this can only be done when we will have the final text. 

Point 4: What is FAO, FAOSTAT Crops, 2020?

Response 4: FAOSTAT means Food and Agriculture Organization Statistical Database. Following the last update of FAOSTAT (March 2023), the corresponding reference was updated (2021) and replaced with the correct reference number [13] in line 64.

Point 5: SISTAT? Or ISTAT 2009?

Response 5: We regret the error in the sentence (ISTAT and not SISTAT). ISTAT reference was replaced with the correct reference number [14] in lines 71 and 74.

Point 6: Complete name of metric units should be written when first mentioned (Kt, Mt)

Response 6: Following the suggestion of Reviewer 3, the complete names of metric units have been added (Kilotonnes = Kt).

Point 7: Table S1: the table should be self-explanatory. What does the PI mean?

Response 7: Table S1 indicates the number of plants that were collected for each smallholding and family garden identified with the name of the keeper farmer. For this reason, after the number of Farm (F1 to F20) were reported the number of sampled plants in each farm: P1 (one sampled plant), P1-2 (two sampled plants), P1-13 (thirteen sampled plants) etc. To make this concept clearer, the caption of the table has been modified.

Point 8: Number all tables and figures in order as they are mentioned.

Response 8: We checked that all Figures and Tables were correctly inserted in the text according to their progressive numbering, and corrected the error in the text if necessary.

Point 9: Some materials and methods are in the results section and viceversa, example: 3.4. Nutritional and chemical characterization. These are materials and methods. Three plants for each of the three replicates for four different genetic groups, two 743 belonging to the Orte 1 population (GGR4 and GGR5) and the others to the Orte 2 popu-744 lation (GGR10 and GGR11), and nine plants for each of the four landraces/clones used as 745 reference genotypes (three plants per replicate) were selected for the determination of nu-746 tritional value and chemical composition of their primary flower heads.

Response 9: As stated before, we revised the text of of the “materials and methods” and “results and discussion” sections, by eliminating some repetitions and information that is not relevant to the overall objectives of the work. In particular, following the suggestion from the Reviewer 3, the  indicated sentence has been removed.

Round 2

Reviewer 1 Report

If possible, it is better to revise this paper according to the comments of last round.

Author Response

Response to Reviewer 1 Comments

If possible, it is better to revise this paper according to the comments of last round.

General Response: We thank the Reviewer for the comment. In the first round of the revision, we have answered all the questions raised by Reviewer 1 and, where possible, his/her suggestions have been incorporated into the text of the new version of the manuscript. With regard to the general comment, as reported in the previous revision, the text of the manuscript has been significantly reduced in the ‘abstract’, ‘introduction’, ‘materials and methods’ and ‘results and discussion’ sections, where some repetitions and information not relevant to the general aims of the work have been removed. These corrections have been made mainly to make the text less long and complex, in accordance with the suggestions from the Reviewer 1. By contrast, the 'conclusions' section was not changed because, as explained in the response to the previous review, it summarises the main results obtained from the different analyses and explains the importance of recovery and conservation of the landrace.

Reviewer 2 Report

I am satisfied with the answers the authors have provided re the comments i raised for this manuscript ' Recovery, morphological, molecular, and nutritional characterization of the globe artichoke “Carciofo Ortano”, a landrace with a high risk of genetic erosion cultivated in Central Italy'. However, i did not see any new reference listed in the reference list. I kindly leave the Supplementary materials to be part of the main manuscript body to the editorial board to judge on. 

Author Response

Response to Reviewer 2 Comments

I am satisfied with the answers the authors have provided re the comments i raised for this manuscript ' Recovery, morphological, molecular, and nutritional characterization of the globe artichoke “Carciofo Ortano”, a landrace with a high risk of genetic erosion cultivated in Central Italy'. However, i did not see any new reference listed in the reference list. I kindly leave the Supplementary materials to be part of the main manuscript body to the editorial board to judge on.

General Response: We thank the reviewer for the comment. Regarding the reference list, we didn’t need to add any new reference, but we have updated the references of the two consulted databases on world (FAOSTAT) and national (ISTAT) production, to the last access (on April 4, 2023).

Furthermore, to meet the suggestion of Revision 2, we have added Figure S1 as Figure 1 and Figure S5 as Figure 4 in the main text.
